# Meta-analysis on the effects of moderate-intensity exercise intervention on executive functioning in children

**Haixia Wang**[1], **Yifei Yang**[2], **Jinfu Xu**[3]*, **Lin Niu**[4], **Yue Zhang**[5], **JingMei Shi**[6], **Lin Shen**[7]

**1** Basic Courses Department, Hebei Agricultural University, Cangzhou, China, **2** School of Economics and Management, Hebei Agricultural University, Baoding, China, **3** School of Physical Education and Sport Science, Fujian Normal University, Fuzhou, China, **4** Dayu Town People's Government of Fengfeng mining area, Handan, China, **5** Hebei Normal University of Science and Technology, Qinhuangdao, China, **6** Zhejiang University of Science and Technology, Shaoxing, China, **7** Humanities and Management School, Hebei Agricultural University, Cangzhou, China

* xujinfu@fjjxu.edu.cn

**Data Availability Statement:** All relevant data are within the paper and its Supporting Information files.

## Abstract

### Objective

We evaluated the effect of moderate-intensity exercise intervention in children and summarized the optimal exercise intervention program.

### Methods

Five significant databases, namely, Web of Science, PubMed, and China National Knowledge Infrastructure, were searched, and the literature was screened strictly according to the inclusion and exclusion criteria and analyzed using Stata 15.1 software.

### Results

There were 25 studies from 22 articles, with a total of 2118 subjects included in the results. According to the meta-analysis, exercise intervention effectively improved children's working memory [SMD = -1.05, 95% CI (-1.26, -0.84)] and cognitive flexibility [SMD = -0.86, 95% CI (-1.04, -0.69)], with a minor improvement in inhibitory control [SMD = -0.55, 95% CI (-0.68, -0.42)].

### Conclusion

a) Improvements in children's working memory and cognitive flexibility by moderate-intensity exercise interventions reached large effect sizes, and improvements in inhibitory control obtained moderate effect sizes. b) Better improvement in working memory for children aged 10 to 12 years than for children aged 6 to 9 years and better cognitive flexibility for children aged 6 to 9 years than for children aged 10 to 12 years. c) Exercise intervention programs lasting 8 to 12 weeks, 3 to 4 times/week, and 30 min/time are most effective in improving executive function in children.

**Funding:** The author(s) received no specific funding for this work.

**Competing interests:** The authors have declared that no competing interests exist.

## Introduction

Executive function refers to the advanced cognitive processes that coordinate and control various basic cognitive processes when completing complex cognitive tasks, which includes three subfunctions: inhibitory control, working memory, and cognitive flexibility. Inhibitory control is the ability of an individual to resist a strong tendency to do something else by controlling their attention, behavior, thoughts, or emotions [1]. Working memory is the ability of an individual to hold information in mind and mentally process or manipulate it, is the basis for inhibitory control and cognitive flexibility [2]. Cognitive flexibility is the ability of an individual to mentally shift, to consider something from a new or different perspective, and to switch between views to accommodate change [3]. Executive function is the core of children's cognitive ability and emotional and social development. Childhood is in the critical period of cerebral cortex development and an important stage of executive function development. Once the development of executive function is impaired, some aspects may be lost in light cases. As a result, learning disabilities, autism, hyperactivity, depression, and other symptoms may appear in some severe cases. Therefore, it is of practical significance to adopt reasonable methods to improve children's executive function.

Executive function in children is plastic, and moderate-intensity exercise is considered the most cost-effective intervention to promote the development of executive function in children [4], which has been supported by the arousal theory hypothesis and neurophysiological mechanism hypothesis [5, 6]. After a 3-year-long exercise intervention experiment, Donnelly et al. found that elementary school students improved their cognitive ability tests, executive abilities, and physical strength levels [7]. Chaddock et al. conducted a 9-month trial of a moderate-intensity aerobic exercise intervention that yielded decreased levels of activation in the right anterior frontal lobe of the child's brain and improved attention and interference resistance performance [8]. Chen et al. experimented and concluded that the moderate-intensity synchronized rope skipping intervention for eight weeks significantly promoted memory refresh and attentional conversion in deaf children but did not substantially improve the inhibition function [9]. Other studies [10, 11] have concluded that a short moderate-intensity exercise can also improve children's executive function. Furthermore, some scholars have also discussed the effect of exercise of different types on improving children's executive function [12, 13].

A large number of findings have been accumulated on the impact of a moderate-intensity exercise intervention on children's executive function. However, currently, there is a lack of studies on the relationship between different intervention programs (exercise intervention cycle, exercise intervention frequency, duration of single intervention) and the development of children's executive function, and the "dose" effect between exercise intervention and each subfunction of executive function cannot be fully revealed. Therefore, in this paper, meta-analysis was conducted to systematically analyze many studies published domestically and internationally to explore the effects of different exercise intervention programs on children's executive function to provide a reference for making appropriate exercise prescriptions.

## Research method

This research project has been registered on the Prospero Systematic Assessment Registry, registration number: 0000000178437040. This research has been reported under the Preferred Reporting Items for Systematic Reviews and Meta-Analyses (PRISMA) guidelines. It strictly follows the PICOS paradigm for describing the literature and developing criteria for literature search, inclusion, screening, and exclusion [4].

## Literature search

The databases searched included Web of Science, PubMed, China National Knowledge Infrastructure, Wanfang Database, and Chinese Science and Technology Journal Database from 2010.1.1 to 2022.10.1. The Chinese search terms included "运动，锻炼，体力活动，体育锻炼，急性运动，有氧运动，学生，儿童，小学生，学龄儿童，执行功能，抑制控制，工作记忆，认知灵活性，实验研究，随机对照，临床试验"; the English search terms included "exercise, exercises, physical activity, physical activities, physical exercise, physical exercises, acute exercise, acute exercises, aerobic exercise, aerobic exercises, child, children, executive function, inhibitory control, working memory, cognitive flexibility, randomized controlled clinical trial, trial". The search strategy used a combination of subject terms and free words, and references to key documents were manually searched. Taking PubMed as an example, the complete search strategy is shown in S1 File.

## Inclusion and exclusion criteria

**Inclusion criteria.** The inclusion criteria were as follows. (a) Subjects: School-age children aged 6~12 years old (physical health); (b) Intervention measures: Exercise intervention with different exercise methods, cycles, frequency, duration, and exercise elements; (c) Comparison of intervention methods: The experimental group was given a moderate-intensity exercise intervention based on the original lifestyle, and the control group maintained their previous lifestyle without any intervention. According to the ACSM's Guidelines for Exercise Testing and Prescription, this study defined moderate intensity exercise intensity at (220-Age)× (40~80%) with a mean heart rate of 120~140 beats/min [14]; (d) Outcome index: At least one of the following outcome indicators was included: inhibitory control, working memory, and cognitive flexibility, measured uniformly in response time (ms); (e) Experimental design: Randomized control experiments were established, with the complete experimental data reported before and after exercise intervention.

**Exclusion criteria.** The exclusion criteria were as follows. (a) Literature in languages other than Chinese and English, review literature, conference papers, and academic dissertations; (b) Literature with research objects that do not meet the requirements; (c) Literature published repeatedly; (d) Literature with experiments that are not randomized and controlled; (e) Literature with incomplete or inaccessible experimental data; (f) This includes children with any physical or mental illness that may prevent or limit their participation in sports interventions.

## Literature screening and data extraction

A total of 794 articles were obtained after preliminary searching, and 218 were left after duplicate articles were deleted with the literature management software. Then, according to the inclusion and exclusion criteria, two authors independently screened the titles and abstracts of potentially eligible studies, and the full texts were downloaded and then cross-checked. In disagreement, a third researcher decided and finally included 22 studies (Fig 1). Then, data extraction was carried out for the screened literature, and the measurement unit was determined to be unified, if not, verified by a third person. The following data were extracted: first author, year of publication, country, the sample size of the experimental group and control group, age, exercise intensity, method of exercise intervention, exercise intervention cycle, frequency, duration, and outcome indices.

## Literature quality evaluation

The literature quality was evaluated according to the Quality Rating Scale developed by Jadad [15]. The scale includes five items: whether the experimental design was described explicitly in

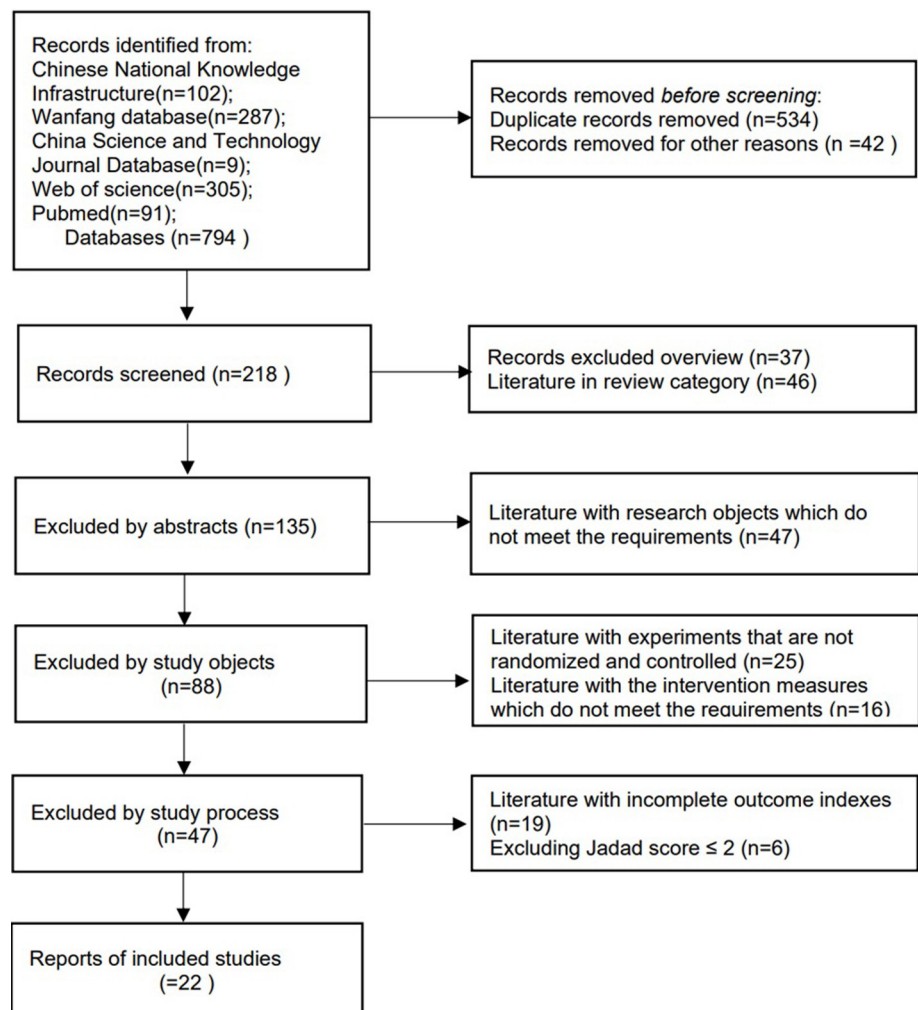

**Fig 1. Inclusion process of the literature search.**

the literature as a randomized controlled trial; whether the blind measurement was used in the experiments; whether the specific methods of the randomized control experiment were introduced; whether the sample size selected completed the whole process of the experiments; and whether the reasons for the subjects dropping out of the study were described. The total score was 5 points, 1 point for conforming to a question item, 0 points for not working, 1 to 2 for low quality, and 3 to 5 for high quality. Two researchers conducted the quality scoring of the literature independently and resolved disagreements by consensus. After evaluation, five studies scored 5, 8 scored 4, and 9 scored 3. All 22 included papers scored three or more, and the quality of the literature was high.

## Data analysis

Merged effect size, heterogeneity test, subgroup analysis and sensitivity analysis were performed on the data extracted from the literature with Stata15.1 software. Since the outcome indices of the literature included in this study were continuous variables, the standard mean difference (SMD) was used as the effect size index. According to Cohen's criteria for evaluating the effect size, $SMD \geq 0.8$ indicated a large effect size, $0.5 \leq SMD < 0.8$ indicated a medium effect

size, 0.2≤SMD<0.5 indicated a small effect size and SMD<0.2 indicated a small effect size [16]. As an index to determine whether there was any heterogeneity between the studies, the range of $I^2$ was from 0% to 100%, and the larger $I^2$ was, the more significant the heterogeneity was. $I^2$<50% or p≥0.1 indicated low heterogeneity between studies, and a fixed effect model was adopted to merge the effect size; $I^2$≥50% or p<0.1 indicated high heterogeneity between studies, a random effect model was adopted to combine the effect size, and subgroup analysis was performed. The funnel plot examination of publication bias was used in this research, and sensitivity analysis was performed with excluded documents per article method.

## Results

### Basic characteristics of the included studies

There were 22 studies from CSCD, SCI, and CSSCI. Of these, two groups were tested in 3 papers (Yin et al. 2014 [14], Yan et al. 2014 [12], and Li et al. 2022 [29]), containing a total of 25 studies. Participants in the 25 studies were from five countries, China, Italy, Germany, the United Kingdom, and the Netherlands, and a total of 2118 study subjects (1086 in the experimental group and 1032 in the control group) were included, with subjects ranging in age from 6 to 12 years.

**Exercise intervention program.** The control group only attended weekly physical education classes and did not engage in any other continuous physical activity; the experimental group added only moderate-intensity exercise intervention to the regular physical education classes. Of the 25 included studies, 8 used short-term exercise interventions, and 17 used long-term exercise interventions. The intervention period ranged from 8 to 24 weeks, the intervention frequency ranged from 2 to 5 times/week, and the intervention duration ranged from 15 to 60 min/time. As shown in Table 1 for details of the research literature.

### Effect of an exercise intervention on executive function

**Meta-analysis of the effect of an exercise intervention on inhibitory control.** Of the 22 included studies, 18 reported the effects of moderate-intensity exercise interventions on children's inhibitory control, with two groups tested in 1 study (Yin et al. 2014 [14]), for a total of 19 studies containing a sample size of 1472 individuals (752 in the experimental group and 720 in the control group). The forest plot of the effect of an exercise intervention on inhibitory control intervention (Fig 2) shows that the heterogeneity test results $I^2$ = 30.2%<50%, p = 0.105>0.1, the heterogeneity between studies is low, and the fixed effects model should be selected for analysis. The combined effect size was SMD = -0.55, 95% CI (-0.68, -0.42), z = 8.25, p<0.01, and the results were statistically significant, indicating that moderate-intensity exercise intervention improved inhibitory control in children.

**Meta-analysis of the effect of an exercise intervention on working memory.** Of the 22 included studies, 16 reported the effects of moderate-intensity exercise interventions on children's working memory, with two groups tested in 1 study (Li et al. 2022 [29])), for a total of 17 studies containing a sample size of 985 individuals (507 in the experimental group and 478 in the control group). The heterogeneity test results showed (Fig 3) that $I^2$ = 55.4%>50%, p = 0.003<0.1, the heterogeneity between studies was high, and a random effects model was selected for analysis. The combined effect size was SMD = -1.05, 95% CI (1.26, -0.84), z = 9.9, p<0.01, and the results were statistically significant, indicating that the moderate-intensity exercise intervention was effective in improving children's working memory levels.

**Meta-analysis of the effect of an exercise intervention on cognitive flexibility.** Of the 22 included studies, 16 reported the effects of moderate-intensity exercise interventions on children's cognitive flexibility, with multiple groups tested in 1 study (Yin et al. 2014 [14]), for a

**Table 1. Basic description of the literature included in the meta-analysis.**

| Author/Time of Publication | Country | Age/ years old | Sample Size E/C | Way of Exercise Intervention | | Exercise Intensity | Duration/ Frequency/Cycle |
|---|---|---|---|---|---|---|---|
| | | | | experimental group | control group | | |
| Cai et al. [5] | China | 10~12 | 24/24 | Complete rope skipping 30 times/group × 20 groups, with a rhythm of 1.5 times/s and an interval of 1min between groups | General Learning Lifestyle | (220-Age) × (60–70%) 125–135 times/min | 50 min, 1 time, 1 day |
| Chen et al. [9] | China | 11~12 | 18/15 | Learn, practice and compete in synchronized rope skipping, supplemented by a few sports games | General Learning Lifestyle | (220-Age) × (60–69%) | 40 min, 3 times/ week, 8 weeks |
| Aiguo et al. [10] | China | 10~11 | 17/17 | Jogging | General Learning Lifestyle | (220- Age) × (60–70%) | 30 min, 1 time, 1 day |
| Chen et al. [11] | China | 9 | 30/30 | High dribble in the same place, jogging and dribbling in progress | General Learning Lifestyle | (220- Age) × (60–69%) | 30 min, 1 time, 1 day |
| Yan et al. (a) [12] | China | 9 | 40/40 | Aerobics step combination and transformation exercise | Weekly physical education classes | (220- Age) × (60–69%) 130–140 times/min | 30 min, 1 time, 1 day |
| Yan et al. (b) [12] | China | 9 | 40/40 | Strength exercises, including squatting, sit-ups and heel lifting | Weekly physical education classes | (220-Age) × (60–69%) 130–140 次/min | 30 min, 1 time, 1 day |
| Alesi et al. [13] | Italy | 8~9 | 24/24 | Football exercise + games | Weekly physical education classes | (220-Age) × (60–70%) | 60 min, 2 times/ week, 24 weeks |
| Yin et al. (a) [14] | China | 10~12 | 109/108 | Martial arts + rope skipping + 8-word running | Weekly physical education classes | (220-Age) × (40–80%) 120–140 times/min | 30 min, 3 times/ week, 10 weeks |
| Yin et al. (b) [14] | China | 10~12 | 109/108 | Martial arts + rope skipping + 8-word running | Weekly physical education classes | (220-Age) × (40–80%) 120–140 times/min | 30 min, 3 times/ week, 20 weeks |
| Yin et al. (b) [17] | China | 10~12 | 62/44 | Learn basketball skills | General Learning Lifestyle | (220-Age) × (60–69%) | 30 min, 3times/ week, 16 weeks |
| Marion et al. [18] | Germany | 6 | 48/53 | Five two-handed exercises include throwing/ kicking balls and hitting the target with left and right hands/feet | Weekly physical education classes | (220-Age) × (60–70%) | 20 min, 1 time, 1 day |
| Yang et al. [19] | China | 10~12 | 20/20 | Campus directional sports | Weekly physical education classes | Moderate intensity 120–140 times/min | 30 min, 3 times/ week, 10 weeks |
| Pan et al. [20] | China | 12 | 48/45 | Basketball technical teaching competition, supplemented by a few sports games | Weekly physical education classes | (220-Age) × (60–69%) | 30 min, 3 times/ week, 10 weeks |
| Chen et al. [21] | China | 11 | 21/20 | Study, practice and competition of football, supplemented by a small number of sports games | Weekly physical education classes | (220-Age) × (60–69%) | 40 min, 2 times/ week, 8 weeks |
| Dai. [22] | China | 11~12 | 46/43 | Techniques, tactical drills, and simulation matches of football | General Learning Lifestyle | (220-Age) × (60–69%) 125–140 times/min | 60 min, 10 times/ week, 24 weeks |
| Liang et al. [23] | China | 11 | 10/10 | Physical and mental exercise | Weekly physical education classes | (220-Age) × (60–70%) | 40 min, 3 times/ week, 8 weeks |
| Xiong et al. [24] | China | 11 | 14/12 | Fancy rope skipping + martial arts exercise + fancy running | General Learning Lifestyle | (220-Age) × (60–69%) | 45 min, 4 times/ week, 11 weeks |
| Wang et al. [25] | China | 6 | 32/31 | Basic standing posture, fist, leg and combination routine of Taekwondo | Weekly physical education classes | Moderate intensity | 45 min, 2 times/ week, 16 weeks |
| Ma et al. [26] | China | 9 | 40/40 | Campus Soccer Training | Weekly physical education classes | Moderate intensity | 40min, 3 times/ week, 16 weeks |
| Vear et al. [27] | Netherlands | 11 | 254/236 | Dance Training | General Learning Lifestyle | Moderate intensity | 15min, 1 time/ week, 9 weeks |
| Khawla et al. [28] | England | 11 | 19/22 | Aerobic dance training | General Learning Lifestyle | Moderate intensity | 45min, 2 times/ week, 8 weeks |
| Li et al. (a) [29] | China | 10 | 20/20 | Basketball marching dribbling + jump rope | Weekly physical education classes | (220-Age) × (60–69%) | 20min, 1 time, 1 day |

(*Continued*)

**Table 1.** (Continued)

| Author/Time of Publication | Country | Age/ years old | Sample Size E/C | Way of Exercise Intervention | | Exercise Intensity | Duration/ Frequency/Cycle |
|---|---|---|---|---|---|---|---|
| | | | | experimental group | control group | | |
| Li et al. (b) [29] | China | 10 | 25/20 | Basketball marching dribbling + jump rope | Weekly physical education classes | (220-Age) × (60–69%) | 40min, 1 time, 1 day |
| Chen et al. [30] | China | 11~12 | 25/20 | Fun Games + Fancy rope skipping + fancy running | Weekly physical education classes | (220-Age) × (60–69%) | 30min, 3 times/ week, 10 weeks |
| Jin et al. [31] | China | 11 | 14/12 | Fancy rope skipping + martial arts exercise + fancy running | Weekly physical education classes | (220-Age) × (60–69%) | 30min, 4 times/ week, 11 weeks |

Note: E: Experimental group; C: Control group.

total of 17 studies containing a sample size of 1704 individuals (859 in the experimental group and 845 in the control group). The heterogeneity test results yielded (Fig 4) $I^2 = 58.3\% > 50\%$, p = 0.001<0.1, with high heterogeneity among studies, and the random effects model was used for analysis.

## Sensitivity analysis and bias test

To explore sources of heterogeneity, first, sensitivity analyses were conducted by excluding literature on a case-by-case basis to assess the effects of each study on indicators of inhibitory control, working memory, and cognitive flexibility. Test results showed that phasing out a particular piece of the study did not significantly change the effect sizes of the three indicators, indicating that the meta-analysis results had some credibility and stability (Fig 5). Next, funnel

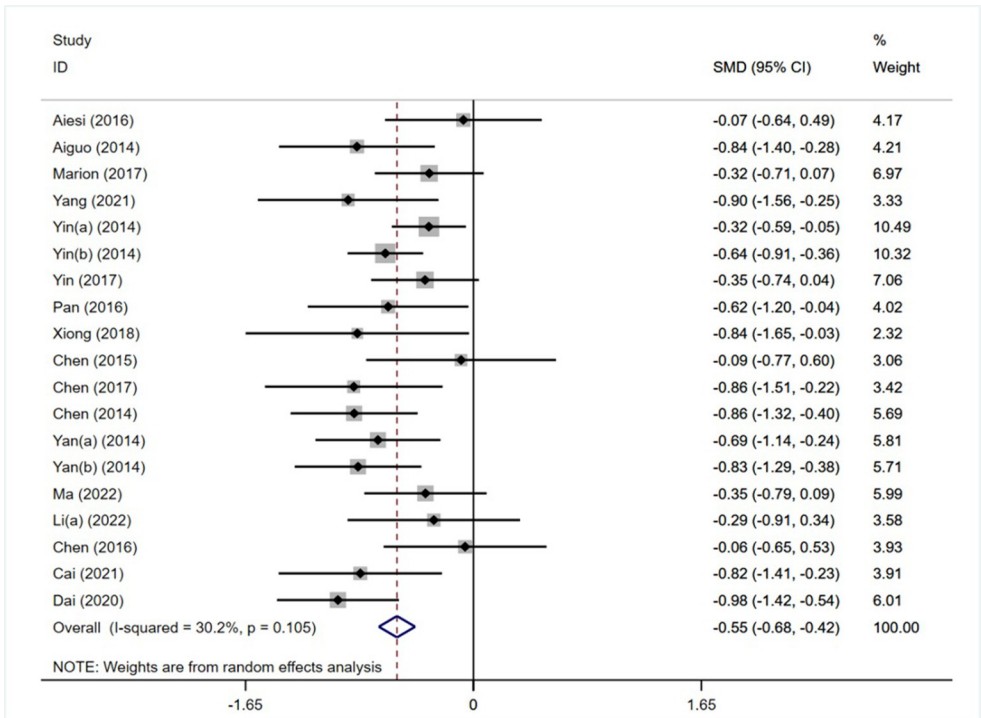

**Fig 2. Forest plot of the effect of motor intervention on inhibitory control.**

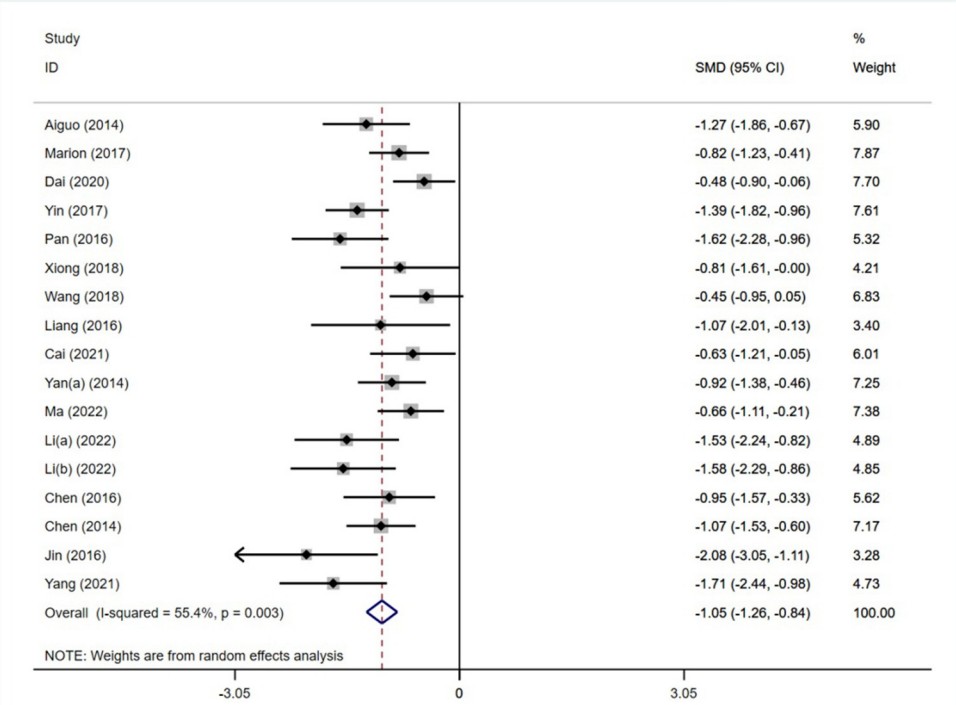

**Fig 3. Forest plot of the effect of an exercise intervention on working memory.**

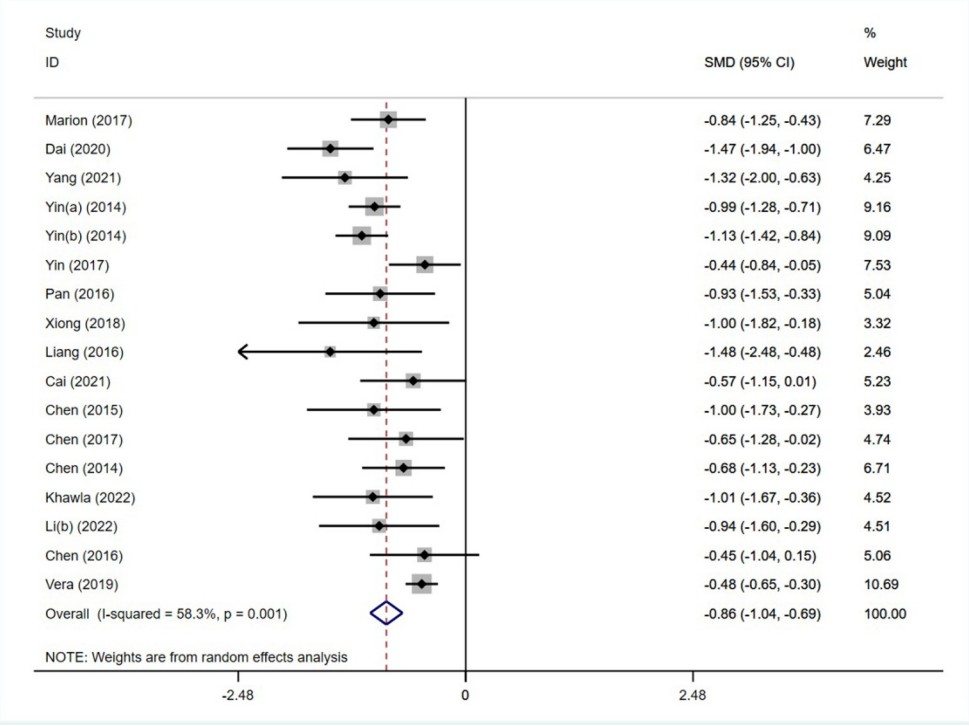

**Fig 4. Forest plot of the effect of an exercise intervention on cognitive flexibility.**

plots were drawn separately for publication bias tests according to inhibitory control, working memory, and cognitive flexibility, with the following results.

Fig 6 shows that the funnel plots of the three subfunctions of the executive function are symmetrical along the central axis, while Egger's bias test yielded p-values greater than 0.05 for all of them; therefore, it can be judged that there is no publication bias in the literature of this study.

## Subgroup moderation effect test

To explore a relatively good exercise intervention scheme, two higher heterogeneity indicators of work memory and cognitive flexibility were subjected to subgroup analysis. Therefore, based on the current results and the characteristics of the included literature, this study set up separate subgroups for analysis in four aspects: age of subjects, intervention cycle, intervention frequency, and intervention time.

**Working memory.** The results of the subgroup analysis of working memory are shown in Table 2.

a. Age of subjects. The 17 RCTs were divided into two subgroups according to the age of the subjects. First, the results showed low heterogeneity ($I^2 = 8.3\% < 50\%$, p = 0.36>0.1) and significant differences (z = 7.51, p<0.01) in the 6~9 years subgroup, with a combined effect size of -0.73, reaching a moderate effect size. Second, heterogeneity was low ($I^2 = 27.1\% < 50\%$, p = 0.19>0.1) and significantly different (z = 10.76, p<0.01) in the 10–12 years subgroup, with a combined effect size of -1.30, reaching a large effect size. The motor intervention improved working memory better in children aged 10 to 12 years than in those aged 6 to 9 years.

b. Exercise intervention cycle. The exercise intervention cycles were divided into three subgroups: 1 time, 8 to 12 weeks, and 16 to 24 weeks. The test $I^2$ for heterogeneity within the three subgroups was 23.6% (p = 0.25>0.1), 19.4% (p = 0.28>0.1), and 0% (p = 0.79>0.1), respectively, with low heterogeneity. The intervention group of 8 to 12 weeks reached the maximum effect size SMD = -1.36, p<0.01, on improving working memory, followed by the 1-time intervention group (SMD = -1.05, p<0.01), which reached a large effect size. In contrast, the lowest improvement in working memory was observed in the 16~24 weeks intervention group (SMD = -0.53, p<0.01).

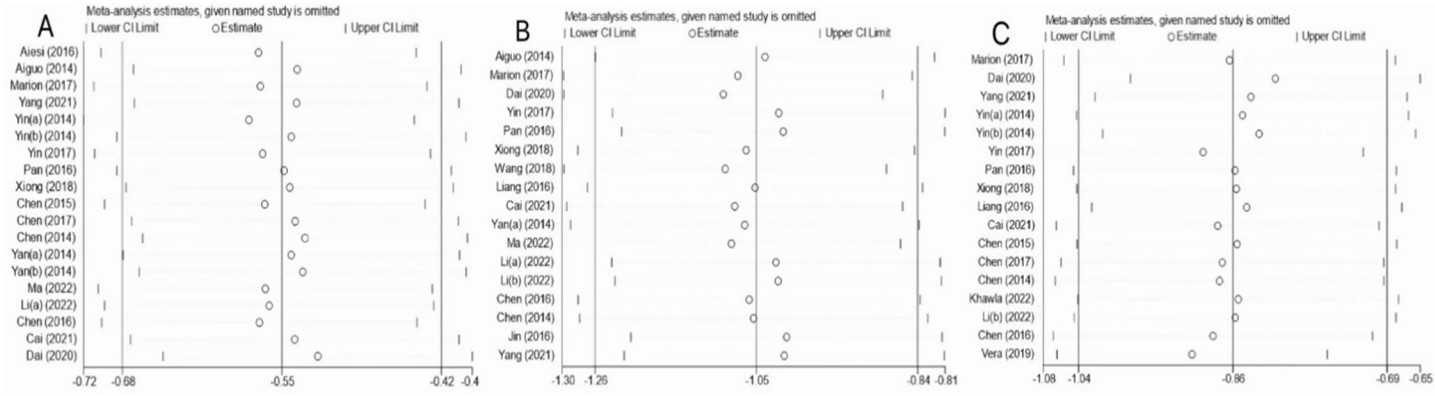

**Fig 5. Sensitivity analysis graph.** Note: A: inhibitory control; B: working memory; C: cognitive flexibility.

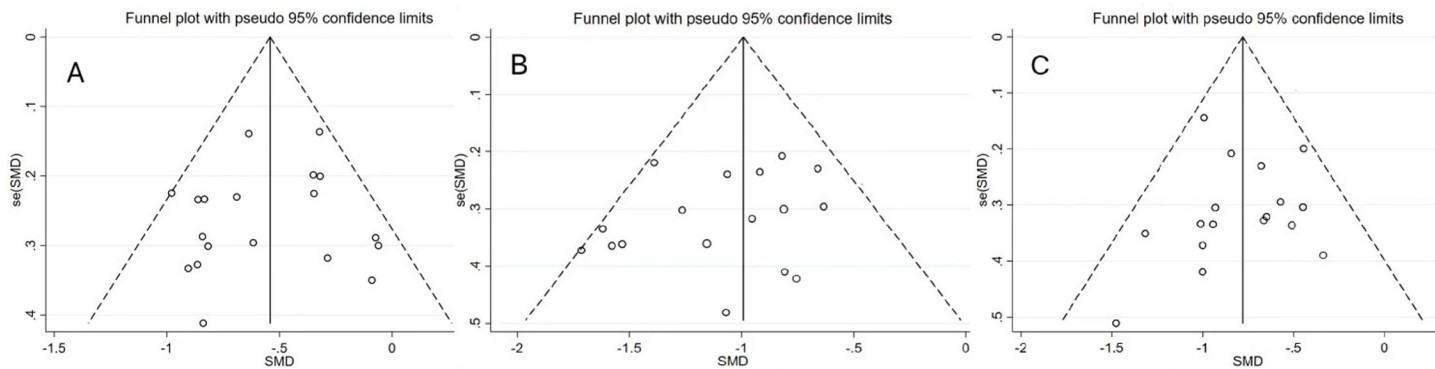

**Fig 6. Publication bias test funnel plots.** Note: A: inhibitory control; B: working memory; C: cognitive flexibility.

c. Exercise intervention frequency. The frequency of exercise intervention was divided into three subgroups: 1 time, 3~4 times/week, and 5~7 times/week. Because only 1 RCT was included in the 5–7 times/week intervention group and no heterogeneity test could be performed, only two subgroups of 1 time and 3–4 times/week were tested for heterogeneity in this study. The test $I^2$ for heterogeneity was 43.1% (p = 0.09<0.1) and 51.6% (p = 0.04<0.1) for the two subgroups, respectively, with low heterogeneity. The 3- to 4-times/week intervention group reached the maximum effect size SMD = -1.24, p<0.01, on improving working memory; the second group was the 1-time intervention group (SMD = -0.98, p<0.01); the lowest improvement in working memory was observed when the frequency of exercise was increased to 5~7 times per week (SMD = -0.48, p<0.01), which may be biased, as only one article was included in this subgroup.

d. Exercise intervention time. The exercise intervention duration was divided into three subgroups: 1 time, 30 min/time, and 40~60 min/time. The three subgroup heterogeneity tests $I^2$ were 31.5% (p = 0.21>0.1), 29.1% (p = 0.22>0.1), and 50.5% (p = 0.07<0.1), respectively, with low heterogeneity. The 30 min/session intervention group reached the maximum effect size SMD = -1.27, p<0.01, in terms of improving working memory; they were followed by the 1-time intervention group (SMD = -1.13, p<0.01), which reached a large

**Table 2. Effect size of stratified subgroup moderating variables on working memory.**

| Variables | Subground | RCT | Sample Size | Heterogeneity test | | SMD | 95%CI | Two-tailed test | |
|---|---|---|---|---|---|---|---|---|---|
| | | | | I²(%) | p | | | z | p |
| Age | 6~9 | 6 | 493 | 8.3 | 0.36 | -0.73 | (-0.92, -0.54) | 7.51 | 0.000 |
| | 10~12 | 11 | 492 | 27.1 | 0.19 | -1.30 | (-1.54, -1.06) | 10.76 | 0.000 |
| Intervention cycle | 1 time | 7 | 442 | 23.6 | 0.25 | -1.05 | (-1.28, -0.81) | 8.81 | 0.000 |
| | 8~12 weeks | 7 | 311 | 19.4 | 0.28 | -1.36 | (-1.65, -1.07) | 9.23 | 0.000 |
| | 16~24 weeks | 3 | 232 | 0 | 0.79 | -0.53 | (-0.80, -0.27) | 3.98 | 0.000 |
| Intervention frequency | 1 time | 8 | 505 | 43.1 | 0.09 | -0.98 | (-1.23, -0.73) | 7.62 | 0.000 |
| | 3~4 times/week | 8 | 391 | 51.6 | 0.04 | -1.24 | (-1.57, -0.91) | 7.30 | 0.000 |
| | 5~7 times/week | 1 | 89 | - | - | -0.48 | (-0.90, -0.06) | 2.23 | 0.026 |
| Intervention time | 1time | 4 | 314 | 31.5 | 0.21 | -1.13 | (-1.42, -0.83) | 7.42 | 0.000 |
| | 30 min/time | 6 | 331 | 29.1 | 0.22 | -1.27 | (-1.56, -0.97) | 8.42 | 0.000 |
| | 40~60 min/time | 7 | 340 | 50.5 | 0.07 | -0.75 | (-1.07, -0.42) | 9.90 | 0.000 |

Note: "-" indicates that no heterogeneity test was performed.

effect size. In contrast, the lowest improvement in working memory was observed in the 40~60 min/session intervention group (SMD = -0.75, p<0.01).

In summary, the moderate-intensity exercise intervention improved working memory in children 10 to 12 years of age more than in children 6 to 9 years of age. In addition, the improvements in working memory from only short-term exercise sessions in the exercise intervention program reached large effect sizes; in long-term exercise, an exercise program in the form of an intervention of 8 to 12 weeks, 3 to 4 times/week, and 30 min/time was most effective in improving working memory in children.

**Cognitive flexibility.** The results of the subgroup analysis of cognitive flexibility are shown in Table 3.

a. Age of subjects. The 17 RCTs were divided into two subgroups according to the age of the subjects. The results showed low heterogeneity ($I^2$ = 40.8%<50%, p = 0.13>0.1) and significant differences (z = 9.31, p<0.01) in the 6–9 years subgroup, with a combined effect size of -0.99, reaching a large effect size. Second, the 10~12 years subgroup was less heterogeneous ($I^2$ = 35.8%<50%, p = 0.11>0.1) and significantly different (z = 7.13, p<0.01), with a combined effect size of -0.74, reaching a moderate effect size. Motor interventions are more effective in improving cognitive flexibility in children aged 6 to 9 years than in those aged 10 to 12 years.

b. Exercise intervention cycle. The exercise intervention cycles were divided into three subgroups: 1 time, 8 to 12 weeks, and 16 to 24 weeks. The test $I^2$ for heterogeneity was 0% (p = 0.80>0.1), 51.1% (p = 0.03<0.1), and 32.6% (p = 0.22>0.1) for the three subgroups, respectively, with low heterogeneity. Among them, the 16~24 weeks intervention group reached the maximum effect size SMD = -0.86, p<0.01 in improving cognitive flexibility, followed by the 8~12 weeks intervention group with a moderate effect size SMD = -0.79, p<0.01 in improving cognitive flexibility, while the 1-time intervention group had the slightest effect size SMD = -0.76, p<0.01.

c. Exercise intervention frequency. The frequency of exercise intervention was divided into three subgroups: 1 time, 3~4 times/week, and 5~7 times/week. The test $I^2$ for heterogeneity was 0% (p = 0.43>0.1) and 38.2% (p = 0.11>0.1) for the two subgroups of 1 and 3 to 4 times/week, respectively, with low heterogeneity. The 5~7 sessions/week intervention group reached the maximum effect size SMD = -1.47, p<0.01, regarding improved

**Table 3. Effect size of stratified subgroup moderating variables on cognitive flexibility.**

| Variables | Subground | RCT | Sample Size | Heterogeneity test | | SMD | 95%CI | Two-tailed test | |
|---|---|---|---|---|---|---|---|---|---|
| | | | | $I^2$(%) | p | | | z | p |
| Age | 6~9 | 6 | 745 | 40.8 | 0.13 | -0.99 | (-1.20, -0.78) | 9.31 | 0.000 |
| | 10~12 | 11 | 959 | 35.8 | 0.11 | -0.74 | (-0.95, -0.54) | 7.13 | 0.000 |
| Intervention cycle | 1 time | 4 | 269 | 0 | 0.80 | -0.76 | (-1.01, -0.51) | 5.98 | 0.000 |
| | 8~12 weeks | 11 | 1129 | 51.1 | 0.03 | -0.79 | (-1.01, -0.58) | 7.27 | 0.000 |
| | 16~24 weeks | 2 | 306 | 32.6 | 0.22 | -0.86 | (-1.04, -0.69) | 7.68 | 0.000 |
| Intervention frequency | 1 time | 7 | 863 | 0 | 0.43 | -0.60 | (-0.73, -0.46) | 8.52 | 0.000 |
| | 3~4 times/week | 9 | 752 | 38.2 | 0.11 | -0.92 | (-1.14, -0.71) | 8.41 | 0.000 |
| | 5~7 times/week | 1 | 89 | - | - | -1.47 | (-1.94, -1.00) | 6.15 | 0.000 |
| Intervention time | 1 time | 4 | 269 | 0 | 0.80 | -0.76 | (-1.01, -0.51) | 5.98 | 0.000 |
| | 15 min/time | 1 | 512 | - | - | -0.48 | (-0.65, -0.30) | 5.34 | 0.000 |
| | 30 min/time | 5 | 614 | 50.2 | 0.09 | -0.80 | (-1.10, -0.51) | 5.37 | 0.000 |
| | 40~60 min/time | 7 | 309 | 39.5 | 0.13 | -1.04 | (-1.36, -0.72) | 6.40 | 0.000 |

cognitive flexibility. However, there may be a significant bias due to the inclusion of only one article. Followed by a 3 to 4 times/weekly intervention group, the high-effect SMD = -0.92, P <0.01, and the minimum improvement effect of the 1-time intervention group in the intervention group, P <0.01.

d. Exercise intervention time. The exercise intervention duration was divided into four subgroups: 1 time, 15 min/time, 30 min/time, and 40~60 min/time. The test $I^2$ for heterogeneity was 0% (p = 0.80>0.1), 50.2% (p = 0.09<0.1), and 39.5% (p = 0.13>0.1) for the three subgroups of 1 time, 30 min/time, and 40–60 min/time, respectively, with low heterogeneity. Among them, the 40–60 min/intervention group reached the most significant effect size in improving cognitive flexibility SMD = -1.04, p<0.1, followed by the 30 min/intervention group with a large effect size SMD = -0.80, p<0.01; the 1-time intervention group had a moderate effect size with an improvement SMD = -0.76, p<0.01; and the 15 min/intervention group had the lowest effect on the improvement of cognitive flexibility (SMD = -0.48, p<0.01).

In summary, the moderate-intensity exercise intervention improved cognitive flexibility in children 6 to 9 years of age more than in those aged 10 to 12. In addition, the improvement in cognitive flexibility with only short-term exercise sessions in the exercise intervention program achieved moderate effect sizes; in long-term exercise, an exercise program in the form of 16~24 weeks, 3~4 times/week, and 40~60 min/time intervention was most effective in improving cognitive flexibility in children.

## Discussion

### Literature quality overall effect size analysis

The results of the Jadad quality score showed that the highest study quality score was 5, and the lowest score was 3. The overall quality of the included literature was high. The results of funnel plots showed that the included literature was evenly separated along the central axis, indicating no bias in the literature. The sensitivity analysis showed a negligible effect on the three subfunctions, indicating reliable analysis results.

The meta-analysis results earlier show that moderate-intensity exercise intervention can improve children's executive functions. Nevertheless, the impact on each subfunction is different, and there is selectivity, which further verifies the research results of predecessors [32]. In this study, the effect of a moderate-intensity exercise intervention on work memory had the largest SMD = -1.05, and the effect of the impacts of cognitive flexibility was second = -0.86. The minor effect on the results of inhibitory control was SMD = -0.55.

The study holds that the brain's frontal lobes, including the motor cortex, premotor cortex, and prefrontal cortex, dominate executive function. The brain area associated with inhibitory control is located in the right inferior frontal gyrus, that associated with working memory is located in the left premotor cortex, and that associated with cognitive flexibility is located in the bilateral frontal cortex, which may influence the neural basis of the effect of motor interventions on the various subfunctions of executive function [33]. During the experiment, different exercise programs stimulate different brain areas, affecting subfunctions differently [33]. To explore the best exercise intervention program, a subgroup analysis was conducted by introducing moderating variables.

### The impact of exercise intervention program on executive function

a. Age of subjects. Subgroup analyses found that the exercise intervention was more effective in improving working memory in children aged 10 to 12 years than in those aged 6 to 9

years, while it was more effective in improving cognitive flexibility in children aged 6 to 9 years than in those aged 10 to 12 years. According to the analysis, the development trajectories of the three subfunctions of executive function were different, among which inhibitory control was the fastest, followed by cognitive flexibility and working memory [34]. First, studies on the development of inhibitory control suggest that inhibitory control develops the fastest at the preschool stage, especially in children aged between 5~8 years old. The improvement space of inhibition function for children over 8 years old is minimal, indicating a significant decline at the end of teenagers, but the accuracy and speed of the execution ability test are significantly improved [35]. The results of this study showed that motor intervention improved the level of inhibitory control in children aged 6 to 12 years with low heterogeneity between studies, which is consistent with the developmental pattern of inhibitory control function. Second, studies on the development of working memory suggest that it is age-dependent and that preschoolers can keep information in their minds for a short time. As children grow older, their brain-derived neurotrophic factor, insulin-like growth Factor 1, and growth hormone gradually increase. The volume and neural network structure of brain-related brain areas will also gradually improve. After the age of 9, children's complex working memory function has steadily improved, entering the first stage of development, and the second development period will last until approximate adolescence [36]. Thus, the improvement effect of exercise on working memory in children aged 10~12 years old (SMD = -1.02) was greater than that in children aged 6~9 years old (SMD = -0.72). Finally, studies on the development of cognitive flexibility suggest that children can complete simple conversion tasks better from 3~4 years old. Being approximately 6 years old is a critical period for the development of children, and from 7 to 9 years old, children enter the rapid growth period of cognitive flexibility. Thus, exercise interventions significantly improve cognitive flexibility in children aged 6 to 9 years. In summary, the development of human executive functions requires a long timeline, especially in the age range of 6 to 12 years. The developmental trajectory of children's executive functions should be noted.

b. Exercise intervention cycle. Exercise intervention effectively promotes the development of executive function in children, but short-term exercise has different effects from long-term exercise. Short-term exercise, also known as episodic exercise, is the primary intention of increasing the inhalation, delivery, and utilization of oxygen in the physical system for a duration of 10 to 60 min. According to the subgroup analysis, 1-time short-term exercise intervention had a specific promotion effect on each subfunction of the executive function, which is consistent with the results of previous studies. Chen et al. conducted a study with fourth graders. They concluded that one 30 min session of moderate-intensity basketball dribbling training was the most effective in improving executive function in children [10]. Davis et al. studied the fMRI method's relationship between short moderate-intensity exercise and cerebral cortical activation in overweight children. The results showed that short moderate-intensity exercise formed positive changes in brain activation patterns for executive function, causing increased bilateral prefrontal activation levels in children when completing functional tasks, indicating that short-term exercise intervention can affect children's executive function and underlying neural networks [37]. Short-term exercise can cause the immediate release of neurotransmitters (dopamine, serotonin), hormones (noradrenaline, growth hormone), and other chemicals; thus, the transient regulatory role that could be exerted by it improves the execution capacity [38]. The analysis showed that the effect of short-term exercise on executive function stems from the inverted U arousal theory hypothesis; in other words, the cognitive level will increase with increased arousal level over a particular range. Other studies have found that the immediate benefit of short moderate-

intensity exercise on executive function can remain for 0.5 h after exercise. However, performance returned to baseline on the three subfunctional tasks after 48 h [39]. The subgroup with 1 intervention in the present study, which was disposable short moderate-intensity exercise lasting for 20~50 min, further confirmed the immediate effect of short-term activity on the improvement of executive function in children.

Subgroup analysis shows that exercise intervention lasting for 8~12 weeks can effectively improve children's executive function. The reason may be that long-term exercise intervention is a process of repetitive practice and constant use of executive functions. Throughout the intervention cycle, the individual's multiple thoughts intersect, create, and imagine so that the experience is constantly reinforced and cognitive functions are improved, which develops executive functions [40]. Research shows that long-term exercise can cause changes in brain structure, resulting in changes in brain cells and molecules that contribute to neuroplasticity. By releasing neurotrophic factors, neurons in the hippocampal gyrus can obtain sufficient nutrients to increase their volume and ultimately strengthen memory and cognitive function [41]. Diamond et al. believed that long-term exercise affects executive function through "motor-cognitive" interconnected brain neural mechanisms [42]. According to functional brain imaging research findings [7], long-term aerobic exercise can enhance cortical functional connectivity in the anterior cingulate, occipital and frontal lobes and improve the overall efficiency of executive function neural networks. Notably, the meta-analysis in this study concluded that when the intervention cycle was greater than 15 weeks, the effect on working memory significantly decreased, possibly due to the impairment of individual self-cognitive tasks and the decreased level of arousal. Therefore, a moderate-intensity exercise intervention for 8~12 weeks is recommended to formulate an exercise program. Nevertheless, from the perspective of the efficacy of exercise for health, persistent exercise over a long time has the same promoting and consolidating effect on the improvement of executive ability.

c. Exercise intervention frequency and exercise intervention time. Subgroup analysis found that exercising 3~4 times per week had the optimal effect on improving executive function. We found that the frequency of intervention plays an essential role in improving the effect of executive function; this is the same as the results of previous research. Yin et al. set up a 10-week experiment with aerobic exercise interventions three times/per week and five times/per week. They concluded that exercising thrice a week significantly enhanced children's executive function. In contrast, exercise 5 times per week only enhanced working memory, and cognitive flexibility had a weaker effect on inhibitory control [14]. In addition, as stated in Health Education and Health Promotion, aerobic exercise at least three times per week is more conducive to developing a student's cognitive level and physical and mental health. According to comprehensive statistics, moderate-intensity exercise intervention 3~4 times per week is recommended to improve executive function.

In terms of exercise intervention time, subgroup analysis revealed that the 30 min/time intervention group maximized children's working memory levels and achieved a large effect size for improved cognitive flexibility. Chen et al. took children in Grade 4 as subjects and set the intervention duration as 8 min, 15 min, and 30 min, respectively. According to the results, performing 30-min aerobic exercises per time improved children's executive function [11]. Davis et al. conducted exercise intervention for 10~15 weeks on 94 obese children and compared the effects of 40 min/d and 20 min/d interventions on executive function, showing that the 40 min/d intervention group scored better than the 20 min/d intervention group in the executive function test task [43]. We found that the improvement effect of executive function is closely related to the duration of each intervention. The relationship between arousal and executive

function is currently described as an inverted U-shaped function. In a specific range, the longer the exercise duration, the more pronounced the improvement effect on executive function. However, after long-term exercise, fatigue and dehydration during exercise will result in individual attention shifting from the current task to the feeling of physical discomfort, which will have adverse effects on cognitive and executive functions. However, the specific "inflection point phenomenon of optimal effect" requires further study [44]. Therefore, it is essential to maintain a reasonable amount of time to maximize exercise's benefits on children's executive function. Combined with the above studies, this study recommends performing moderate-intensity exercise for 30 minutes, effectively improving children's executive function.

## Limitations and outlook

Although this study strictly followed the meta-analysis process, there are some limitations. First, in terms of moderating variables, future studies can be conducted after increasing exercise locations, different exercise intensities, and different genders to improve exercise prescription. Second, the number of studies included was limited, and more studies can further explore practical ways to improve children's executive function in the future.

## Conclusions

This article applied literature selection with strict inclusion and exclusion criteria by searching five significant databases, including Web of Science, PubMed, and China National Knowledge Infrastructure, and analysis using Stata15.1 software. The effectiveness of moderate-intensity exercise interventions in improving executive function in children was evaluated, and the optimal exercise intervention program was summarized as follows.

a. Moderate-intensity exercise intervention improves executive function in children, which has more significant effects on working memory and cognitive flexibility but less on inhibitory control. b) The moderate-intensity exercise intervention was better for improving working memory in 10- to 12-year-old children than in 6- to 9-year-olds and better for improving cognitive flexibility in 6- to 9-year-olds than in 10- to 12-year-olds. c) Exercise intervention programs lasting 8 to 12 weeks, 3 to 4 times/week, and 30 min/time are most effective in improving executive function in children.

## Supporting information

**S1 File. In this study, PubMed and China National Knowledge Infrastructure were used as examples, and the complete search strategy is shown in Tables 1 and 2.**
(DOC)

**S1 Checklist. PRISMA 2020 checklist.**
(DOCX)

**S2 Checklist. PRISMA 2020 flow diagram for new systematic reviews which included searches of databases and registers only.**
(DOCX)

## Author Contributions

**Data curation:** Yifei Yang.

**Formal analysis:** Jinfu Xu.

**Investigation:** Lin Shen.

**Resources:** Haixia Wang, Yue Zhang, Lin Shen.

**Software:** Haixia Wang, Lin Niu, JingMei Shi.

**Writing – original draft:** Haixia Wang.

**Writing – review & editing:** Haixia Wang.

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
