## [Decision Letter · Decision Letter 0]

21 Jun 2022

PONE-D-22-08574Meta-analysis on the Effects of Moderate-intensity Exercise Intervention on Executive Functioning in ChildrenPLOS ONE

Dear Dr. Wang,

Thank you for submitting your manuscript to PLOS ONE. After careful consideration, we feel that it has merit but does not fully meet PLOS ONE’s publication criteria as it currently stands. Therefore, we invite you to submit a revised version of the manuscript that addresses the points raised during the review process.

We look forward to receiving your revised manuscript.

Kind regards,

Andrew Philip Lavender, PhD

Academic Editor

PLOS ONE

Journal Requirements:

Reviewers' comments:

Reviewer's Responses to Questions

**Comments to the Author**

1. Is the manuscript technically sound, and do the data support the conclusions?

Reviewer #1: No

Reviewer #2: Yes

2. Has the statistical analysis been performed appropriately and rigorously? 

Reviewer #1: No

Reviewer #2: Yes

3. Have the authors made all data underlying the findings in their manuscript fully available?

Reviewer #1: Yes

Reviewer #2: Yes

4. Is the manuscript presented in an intelligible fashion and written in standard English?

Reviewer #1: No

Reviewer #2: Yes

5. Review Comments to the Author

Reviewer #1: The conclusions of this manuscript are uninterpretable to me without

1. A Pub Med search

2. Use of PRISMA guidelines

3. A greater emphasis on the strengths of limitations of meta-analysis.

4. A greater emphasis on bias and confounding without undue emphasis on the P values

Reviewer #2: This topic has practical significance, the demonstration process needs to be further improved, and the conclusion needs to be further considered. The following opinions are put forward:

1. Section Literature quality evaluation：“(9 points 5 points, 7 points to get 4 points, 5 articles 3 points)”

Please clarify your meaning.

2. Section Data analysis: “As an index to determine whether there was any heterogeneity between the studies, the range of I2 was from 0% to 100%, and the bigger I2 was, the greater the heterogeneity was. I2<50% indicated a low heterogeneity between studies, a fixed effect model was adopted to merge the effect size; I2≥50% indicated a high heterogeneity between studies, a random effect model was adopted to merge the effect size, and sub-group analysis was performed”

Please add references.

3. Section Basic characteristics of included studies: “A total of 21 literature were included in this study, which were all published in core journals, covering 30 studies. Among them, there were 3 literature (Chen et al.[11], Tomporowski et al. [24], Yin et al.[25], Yan et al.[33]) that included 2 studies, 1 literature (Yan et al.[14]) that included 3 studies, and 1 literature (Yin et al.[19]) that included 4 studies. A total of 2,466 subjects (1,265 in experimental group and 1,201 in control group), aged 6~12 years old, were included.”

1)How to define “core journals”?

2)The language expression is ambiguous, please clarify the relationship “A total of 21 literature” and “covering 30 studies”.

3) “Among them, there were 3 literature (Chen et al.[11], Tomporowski et al. [24], Yin et al.[25], Yan et al.[33]) that included 2 studies,” The author proposed “3 references”, and the results were followed by 4 references, Please check.

4)The author proposed ‘A total of 2,466 subjects’, but inconsistent with the total sample size in Table 1, Please check.

4. Section Effect of exercise intervention on executive function: “indicating a statistically significant increase in refresh function in children after exercise intervention compared to the control group” and “indicating a statistically significant increase in conversion function in children after exercise intervention”.

“after exercise intervention” should be revised to “experimental group”

5. Section Basic characteristics of included studies

The amount of physical activity in the control group should be clearly described.

6. Please check the number of references in Table 1 and Table 4.

7. The World Health Organization has proposed the recommended amount of physical activity for children, which requires that the health benefits of long-term exercise should be better. However, the conclusion of this study recommends that 8-12 weeks of moderate intensity exercise has a good effect on the improvement of executive function, which is in contradiction with the health benefits of long-term exercise. Please draw this conclusion carefully. Please pay attention to this contradiction in the discussion and explain it.

6. PLOS authors have the option to publish the peer review history of their article (what does this mean?). If published, this will include your full peer review and any attached files.

Reviewer #1: No

Reviewer #2: No

---

## [Author Response · Author response to Decision Letter 0]

22 Jul 2022

We have submitted the revised draft as required, including "reply to reviewers, revised manuscript and trajectory changes, manuscript".

---

## [Decision Letter · Decision Letter 1]

26 Sep 2022

PONE-D-22-08574R1Meta-analysis on the Effects of Moderate-intensity Exercise Intervention on Executive Functioning in ChildrenPLOS ONE

Dear Dr. Wang,

Thank you for submitting your manuscript to PLOS ONE. The original editor was not available and I took over the processes. After careful consideration, we feel that it has merit but does not fully meet PLOS ONE’s publication criteria as it currently stands. Therefore, we invite you to submit a revised version of the manuscript that addresses the points raised during the review process.

I read your revision. I can see the importance of your report. I, however, agree with the first reviewer that the quality of the writing is not good and there are many unclear aspects that makes it very hard to understand the content. For exampleyour English search terms included "Primary student" which does not have any particular meaning in English. Or "Medium intensity" which is wrong in English and it has to be "Moderate intensity". These are very important to be clarified. Subsequently you mention that "The search strategy is a combination form of separate and mutual, supplemented by literature tracking." What does this mean? You might want to include your search query for clarification (for both Chinese and English searches); it could be as a supplementary document if it is long. In the first glance I did not understand what you mean by "refresh" or "conversion" function. I noticed you cited two articles [1-2]. I looked at the review article to see if those words were used. I couldn't find these words. You need to use conventional terms that researchers are familiar with. You need to be clearer in terms of executive functions you looked at such as working memory or inhibition.I still believe you have not responded satisfactorily to the comments of the reviewers, in particular reviewer 1. For example, I do not see much discussion on the "location" in the document. In table 1, column Control Group, you mention "regular physical training". What are these? How do these differ with the physical activity in the Experimental Group? There are many grammatical or potentially conceptual errors in the writing. I am not first English language speaker. But I noticed many grammatical errors.

Minor points:

Perhaps you want to change the term "literature" to "publication" or "article" or "study".Sometimes you say "execution" and sometimes you say "executive". Am I right that all the items in Table 4 include an asterisk? If so, then you don't need to have the asterisk. You can simply mention "P<0.00001" in the caption or notes of the figureI am giving you one more chance to revise the document and resubmit. I would suggest you to ask an English speaking person read your document carefully in terms of concept and language. Somebody who is not necessarily in the field to give you feedback on the readability of the text in addition to correct the grammatical errors. I will not send the document to be reviewed again and will make the final decision myself. But I would expect a substantial change. Thanks.

We look forward to receiving your revised manuscript.

Kind regards,

Amir-Homayoun Javadi, PhD

Academic Editor

PLOS ONE

Reviewers' comments:

Reviewer's Responses to Questions

**Comments to the Author**

1. If the authors have adequately addressed your comments raised in a previous round of review and you feel that this manuscript is now acceptable for publication, you may indicate that here to bypass the “Comments to the Author” section, enter your conflict of interest statement in the “Confidential to Editor” section, and submit your "Accept" recommendation.

Reviewer #1: (No Response)

Reviewer #2: All comments have been addressed

2. Is the manuscript technically sound, and do the data support the conclusions?

Reviewer #1: Partly

Reviewer #2: Yes

3. Has the statistical analysis been performed appropriately and rigorously? 

Reviewer #1: No

Reviewer #2: Yes

4. Have the authors made all data underlying the findings in their manuscript fully available?

Reviewer #1: No

Reviewer #2: Yes

5. Is the manuscript presented in an intelligible fashion and written in standard English?

Reviewer #1: No

Reviewer #2: Yes

6. Review Comments to the Author

Reviewer #1: I find it impossible to interpret these results without a clear statement of the type of study design of each component in the meta analysis as well as the results of the individual trials.

I am unclear whether there is effect modification by country in which the individual study was performed.

I also believe that whatever the answers to the above mentioned queries the results should be interpreted as hypothesis generating not testing

Reviewer #2: The author has carefully revised the questions raised, and the quality of the paper has been greatly improved.

7. PLOS authors have the option to publish the peer review history of their article (what does this mean?). If published, this will include your full peer review and any attached files.

Reviewer #1: No

Reviewer #2: No

---

## [Author Response · Author response to Decision Letter 1]

25 Nov 2022

The author has revised and checked the graphs

---

## [Editor Report · Decision Letter 2]

15 Dec 2022

Meta-analysis on the Effects of Moderate-intensity Exercise Intervention on Executive Functioning in Children

PONE-D-22-08574R2

Dear Dr. Xu,

We’re pleased to inform you that your manuscript has been judged scientifically suitable for publication and will be formally accepted for publication once it meets all outstanding technical requirements.

Kind regards,

Amir-Homayoun Javadi, PhD

Academic Editor

PLOS ONE
---

## [Editor Report · Acceptance letter]

31 Jan 2023

PONE-D-22-08574R2 

Meta-analysis on the Effects of Moderate-intensity Exercise Intervention on Executive Functioning in Children 

Dear Dr. Xu:

I'm pleased to inform you that your manuscript has been deemed suitable for publication in PLOS ONE. Congratulations! Your manuscript is now with our production department. 

Kind regards, 

on behalf of

Dr. Amir-Homayoun Javadi 

Academic Editor

PLOS ONE